# Stakeholders' Perceptions of How Nurse–Doctor Communication Impacts Patient Care: A Concept Mapping Study

**Sandesh Pantha** [1,*] **, Martin Jones** [2] **and Richard Gray** [1]

1   School of Nursing and Midwifery, La Trobe University, Bundoora, VIC 3086, Australia; r.gray@latrobe.edu.au
2   Department of Rural Health, University of South Australia, Whyalla Noorie, SA 5608, Australia;
    martin.jones@unisa.edu.au
*   Correspondence: s.pantha@latrobe.edu.au

**Abstract:** There is some evidence that aspects of nurse–doctor communication are associated with the quality of care and treatment patients receive whilst they are in hospital. To date, no studies have examined stakeholder perceptions on how patient care is influenced by clinical communication between nurses and doctors. We conducted a concept mapping study to generate a deep understanding of how clinical communication impacts patient care. Concept mapping has six phases: preparation, idea generation, structuring, representation, interpretation, and utilization. A total of 20 patients, 21 nurses, and 21 doctors participated in the study. Brainstorming generated 69 discreet statements about how nurse–doctor communication impacts patient care. The structuring (rating and clustering) phase was completed by 48 participants. The data interpretation workshop selected a five-cluster solution: effective communication, trust, patient safety, impediments to patient care, and interpersonal skills. On the final concept map, the five clusters were arranged in a circle around the center of the map. Clusters were relatively equal in size, suggesting that each concept makes a broadly equal contribution to how nurse–doctor communication influences patient care. Our study suggests that there are multiple aspects of clinical communication that impact patient care. Candidate interventions to enhance nurse–doctor communication may need to consider the complex nature of interprofessional working. Registration: This study was prospectively registered with the Open Science Framework (OSF) on 09.07.2020 (osf.io/9np8v/) prior to recruiting the first participant.

**Keywords:** nurse–doctor communication; quality; patient care; concept mapping; nursing; medicine

## 1. Introduction

Nurses and doctors spend more time than any other professional groups providing direct patient care [1,2], spending on average between a quarter to half of their practice hours in this activity [3–5]. Around 90% of the care patients receive whilst they are in hospital are from nurses and doctors [1]. There is considerable overlap in the scope of practice between nurses and doctors, underscoring the importance of effective and accurate interdisciplinary communication [6–8]. Patient care may be negatively impacted when nurse–doctor communication is inaccurate, distorted, or delayed [7,8].

Several authors have investigated the association between the quality of nurse–doctor communication and patient outcomes [9–13]. For example, Swiger et al. (2017) conducted a systematic review of 46 studies of nurse–doctor communication, of which 14 were focused on associations with patient outcomes [13]. The narrative synthesis indicated that there was seemingly a consistent association between the quality of communication and reduced rates of medication errors and hospital-acquired pressure ulcers [13]. Kang et al. (2020) examined the association between the quality of nurse–doctor communication, determined using the Practice Environment Scale of the Nursing Work Index (PES-NWI), and 30-day mortality in surgical patients. The study involved 29,391 nurses and 1.32 million patients from 665 acute-care hospitals (Kang et al., 2020). The authors reported that a one-unit

increase in the quality of nurse–doctor communication was associated with a 5% reduction in hospital mortality [12].

Few studies have examined the association between the amount of nurse–doctor communication and patient outcomes [14–17], reporting inconsistent findings. Higgins et al. (1999), for example, reported no association between the amount of nurse–doctor communication and mortality and readmission in 175 patients admitted to Intensive Care. Conversely, Baggs et al. (1992) reported an association between the amount of nurse–doctor communication and patient outcomes. Rothberg et al. (2012) demonstrated an association between the amount of time doctors spend communicating with nurses and levels of agreement on patient care plans. However, there is a paucity of evidence on how nurse–doctor communication impacts patient care. Developing a deeper understanding may inform the development of strategies to enhance interdisciplinary working [18].

## 2. Methods

Concept mapping is a mixed methods design extensively used to develop an understanding of complex problems [19,20]. There are six phases in concept mapping: preparation, idea generation (brainstorming), structuring (clustering and prioritization), representation (generation of concept map), interpretation, and utilization.

A protocol describing the methodology for this study has been previously reported [21]. We provide key details of the six phases of concept mapping that we followed.

### 2.1. Phase 1, Preparation

The aim of the first phase is to determine stakeholder groups that will be involved in the research and establish a focus question for the study. The three stakeholder groups identified were patients, nurses, and medical doctors. The focus question was developed and refined through consultation with representatives of each of the stakeholder groups and members of the research team. We arrived at the focus question "How does nurse-doctor communication impact patient care?".

2.1.1. Stakeholder Groups

For this research, we defined each of the three stakeholder groups as follows: we considered a patient to be anyone over the age of 18 that had been an inpatient in a medical or surgical ward for at least 24 h within the last 12 months. Nurses and doctors were defined as registered health workers that spent at least one full day a week providing direct patient care in any clinical setting.

2.1.2. Recruitment and Consent

We recruited participants by posting information about the research on various social media platforms (Supplementary Document S1 is the social media advert used for the study), including the "Doctors in Australia" and "The Nurse break" social media groups. People interested in taking part in the research were asked to contact the study researcher who sent by email the participant information and consent form and arranged a meeting to explain the study and address any questions that they may have about the research. This conversation included a discussion about the possible risks associated with participating.

Participants willing to take part in the research provided an online electronic consent procedure using REDCap (Research Data Capture). We checked consent with all participants before each phase of concept mapping.

Immediately following the consent procedure, participants were invited to provide basic demographic information (requested information listed in the protocol, Pantha et al. (2021) [21]) by completing an online survey using REDCap.

### 2.2. Phase 2, Statement Generation (Brainstorming)

Statement generation was conducted via video-conferenced individual interviews. The audio component of the interview was retained. Participants were asked to respond

to the focus question, "How does nurse-doctor communication impact patient care?" The researcher would invite participants to elaborate on responses by asking supplementary questions (e.g., can you tell me a bit more about. . .).

Each interview was listened to by a researcher up to three times to identify candidate statements. A statement was considered a sentence or phrase containing a single topic, that was easy to understand, and that did not contain any jargon or acronyms [22]. Statements from all study participants were then combined into a single document, and duplicate or essentially similar statements were merged into a single item. Statements that did not relate to the focus question were removed. Statement reduction continued until there were fewer than 98 (the maximum number that can be analyzed by the concept mapping software we used).

### 2.3. Phase 3, Structuring of Statements

Structuring of statements required participants to complete two tasks—clustering and prioritization—both undertaken using the online concept mapping software package Ariadne (v 3.0) developed by Peter Severans from The Netherlands.

For the clustering task, participants were asked to group statements that seemed to belong together in up to 10 groups. Participants were advised that all statements could not be piled up into a single cluster and there could not be a "miscellaneous" group. Additionally, they were asked to generate a label for each cluster.

During the prioritization task, each statement was ranked on a five-point Likert scale from one (least important) to five (most important). Participants were instructed to ensure that each point on the scale had an equal number (or as close to equal as was possible, if the number of statements was not divisible by five) of statements.

Study participants were sent a link and a step-by-step guide (https://doi.org/10.26181/5f43450ce2999, accessed on 1 August 2023) to complete the structuring tasks. A follow-up email was sent after three days to check if there were any issues or problems completing the tasks or if additional support was required. If participants did not complete the task, up to three reminder emails were sent.

### 2.4. Phase 4, Representation of the Statements

In this phase, a series of candidate concept maps were generated from the clustering data using the "Ariadne" software package. There were three steps in the data analysis: 1. group similarity matrix, 2. principal component analysis, and 3. hierarchical cluster analysis [23]. Clustering data were coded as a binary response ("1" when two statements were grouped and "0" when not grouped) to generate a group similarity matrix. A principal component analysis then transformed the group similarity matrix data into a two-dimensional space, known as a "point map". Each statement is represented as a dot on the map. Statements frequently grouped together appear closer; those infrequently arranged in a pair stay at a distance [24]. The objective of the principal component analysis is to flatten the multidimensional data into a two-dimensional space without the loss of its structural integrity [25]. Finally, a hierarchical cluster analysis was computed to produce a series of maps having between 2 and 18 cluster solutions.

### 2.5. Phase 5, Data Interpretation

Data interpretation was undertaken in four steps. First, the research team reviewed each of the 17 candidate concept maps to eliminate cluster solutions that had overcrowding of statements in one cluster. In addition, concept maps with single statements forming separate clusters were also excluded. Next, we asked participants who completed the structuring phase if they would agree to participate in a single data interpretation workshop. Participants reviewed and discussed each of the candidate cluster solutions to agree on a final concept map that best represented the data. SP reviewed the point map to identify the theme reflected by the statements around the opposite extremes of the $x$ and $y$ axes and proposed labels to the idea captured by the four ends. The research team reviewed

and determined the axis labels. The final step involved calculating an average importance rating for each cluster from the final concept map, overall and separately for each group.

*2.6. Ethical Considerations*

The La Trobe University human ethics committee reviewed and approved the study (HREC approval number 20172, 10 June 2020). The main ethical consideration for this study was the decision to reimburse participants for their time, which may be considered an inducement to participate. Each participant was paid AUD 25 for completing each phase of the study. The value of the payment was consistent with the Australian National Health and Medical Research Council guidance [26].

Potential study participants were sent written information about the study and then asked to provide consent electronically. There was no opportunity for researchers to check potential participants' signing of the consent form. However, the study was considered low-risk, and our procedures for providing information and consent were considered appropriate by the research ethics committee.

Finally, it was possible that talking about the hospital experiences may have been distressing to some participants. Where this occurred, we had in place a stepped approach to provide additional support to participants.

## 3. Results

Fieldwork for the study was carried out between 8 October 2020 and 14 June 2021 (brainstorming interviews, 8 October 2020 through 5 February 2021 and structuring phase, 7 May through 14 June 2021). The data interpretation session was held on 8 September 2021.

Figure 1 shows the flow of participants through the different phases of the study. Eighty-two people expressed an interest in participating. Of 78 who met our inclusion criteria, 68 provided written informed consent to take part in the research.

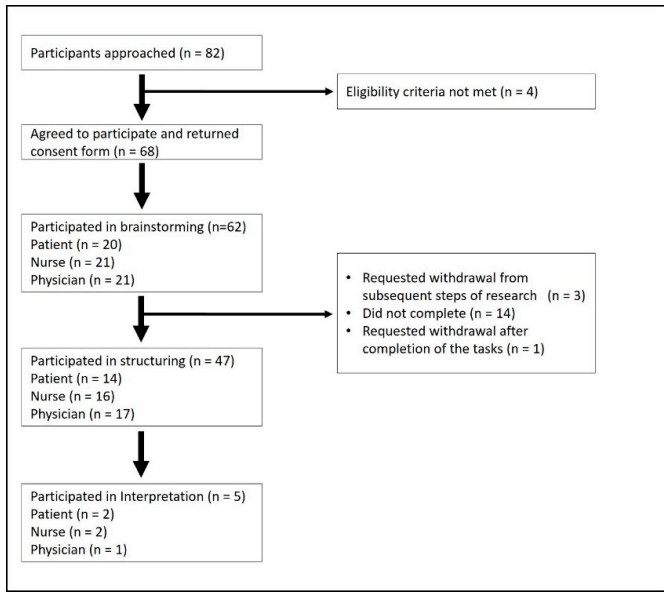

**Figure 1.** Flow of participants through different phases of concept mapping.

Table 1 shows the demographic characteristics of study participants overall and by stakeholder groups. Most participants were female, in their mid-thirties, and born in Australia. Around two-thirds were educated to a post-graduate level. Nurses and doctors who participated were generally trained in Australia. Nurses had almost a decade more clinical experience compared with doctors. There were no apparent systematic differences between participants that took part in the different phases of concept mapping.

**Table 1.** Participants' demographic characteristics.

| Characteristics | | All Brainstorming (*n* = 62) | All Clustering and Prioritization (*n* = 47) | Patient Brainstorming (*n* = 20) | Patient Clustering and Prioritization (*n* = 13) | Nurse Brainstorming (*n* = 21) | Nurse Clustering and Prioritization (*n* = 16) | Doctor Brainstorming (*n* = 21) | Doctor Clustering and Prioritization (*n* = 18) |
|---|---|---|---|---|---|---|---|---|---|
| Gender (Female) | | 47 (77%) | 38 (81%) | 14 (70%) | 10 (77%) | 19 (90.5%) | 14 (88%) | 14 (78%) | 14 (78%) |
| Age in years (Mean, SD) | | 35.9 (10.2) | 38.2 (12.3) | 45 (17) | 43.4 (14.7) | 41 (10.3) | 42.6 (11.1) | 30.3 (6.9) | 30.5 (7.1) |
| Country of Birth | Australia | 34 (55%) | 28 (60%) | 11 (55%) | 8 (62%) | 10 (48%) | 9 (56%) | 13 (62%) | 11 (62%) |
| | Other | 28 (45%) | 19 (40%) | 9 (45%) | 5 (38%) | 11 (52%) | 7 (44%) | 9 (38%) | 7 (38%) |
| Highest educational qualification [1] | Undergraduate | 2 (3%) | 2 (4%) | 2 (10%) | 2 (15%) | - | - | - | - |
| | Graduate | 19 (31%) | 14 (30%) | 5 (25%) | 2 (15%) | 4 (19%) | 3 (19%) | 10 (48%) | 9 (50%) |
| | Postgraduate | 39 (63%) | 29 (62%) | 13 (65%) | 9 (70%) | 16 (76%) | 12 (75%) | 10 (48%) | 8 (44%) |
| Country of clinical qualification | Australia | 32 (76%) | 27 (80%) | - | - | 15 (71%) | 12 (75%) | 17 (81%) | 15 (83%) |
| | Other | 10 (24%) | 7 (20%) | - | - | 6 (29%) | 4 (25%) | 4 (19%) | 3 (17%) |
| Years of clinical work (Mean, SD) [2] | | 9.3 (9.3) | 9.3 (9.9) | - | - | 13.5 (10.4) | 14.6 (11.2) | 4.8 (5.2) | 4.2 (4.6) |
| Years at current workplace (Mean, SD) [2] | | 4.1 (5.2) | 4.3 (5.4) | - | - | 6.3 (6.3) | 7 (6.7) | 2 (1.7) | 1.8 (1.6) |
| Clinical setting | Medical ward | - | - | 7 (35%) | 4 (30%) | - | - | - | - |
| | Surgical ward | - | - | 12 (60%) | 8 (62%) | - | - | - | - |
| | Do not know | - | - | 1 (5%) | 1 (8%) | - | - | - | - |

[1] Information not available for two participants. [2] One of the doctors did not provide information.

### 3.1. Phase 2, Idea Generation

3.1.1. Brainstorming

In total, 62 participants generated 1307 individual statements (a complete list of all statements is included in Supplementary Document S2). On average, brainstorming interviews lasted 26 min.

3.1.2. Statement Reduction

Our first round of statement reduction removed 937 statements (Supplementary Document S3). A further 311 (Supplementary Document S4) statements were removed during the second round. The final list had 69 discreet statements (Supplementary Document S5). Statements were edited for clarity, retaining the original wording as far as possible [24].

### 3.2. Phase 3, Structuring of the Statements

Fifty-two participants started structuring tasks (prioritization and clustering) that were completed by forty-eight. One participant withdrew from the study after completion of both tasks, requesting their data be removed and destroyed. The final data set comprised data from 47 participants who completed both tasks. Participants organized statements into an average of six clusters. The importance rating (mean, standard deviation, and 95% confidence interval) for each statement is shown in Table 2.

**Table 2.** Rating score for sixty-nine statements, disaggregated by stakeholder groups.

| Number [1] | Statement | All Stakeholders | | Patient | | Nurse | | Doctor | |
|---|---|---|---|---|---|---|---|---|---|
| | | Mean, SD, | 95% CI | Mean, SD, | 95% CI | Mean, SD, | 95% CI | Mean, SD, | 95% CI |
| **Cluster 1, Effective Communication** | | **3.4 (1.3)** | **3.1, 3.8** | **3.5 (1.2)** | **2.8, 4.2** | **3.5 (1.3)** | **2.9, 4.2** | **3.4 (1.2)** | **2.8, 3.9** |
| 23 | Precise communication is required in emergency situations (e.g., cardiac arrest) | 4.6 (1.1) | 4.3, 4.9 | 4.9 (0.4) | 4.7, 5.1 | 4.2 (1.5) | 3.5, 5.0 | 4.7 (0.9) | 4.3, 5.1 |
| 61 | Clear and detailed clinical documentation is an important aspect of nurse–doctor communication | 4.2 (1.0) | 4.0, 4.5 | 3.7 (1.3) | 3.0, 4.5 | 4.5 (0.8) | 4.1, 4.9 | 4.3 (0.9) | 3.9, 4.7 |
| 4 | Effective nurse–doctor communication improves the quality of patient care | 4.1 (1.2) | 3.7, 4.4 | 3.8 (1.4) | 3.0, 4.6 | 4.1 (1.4) | 3.4, 4.8 | 4.2 (1.0) | 3.7, 4.7 |
| 13 | Effective nurse–doctor communication ensures timely patient care | 3.9 (1.2) | 3.6, 4.3 | 4.1 (0.9) | 3.6, 4.7 | 3.9 (1.0) | 3.4, 4.4 | 3.8 (1.4) | 3.1, 4.5 |
| 17 | Good communication is important across all shifts (including nights) | 3.9 (1.2) | 3.5, 4.2 | 3.7 (1.3) | 3.0, 4.5 | 3.7 (1.3) | 3.0, 4.3 | 4.1 (1.1) | 3.6, 4.6 |
| 8 | Nurses need ensure they are aware of change in patients' care plans | 3.9 (1.1) | 3.7, 4.2 | 4.0 (1.1) | 3.4, 4.6 | 4.5 (0.6) | 4.2, 4.8 | 3.4 (1.1) | 2.9, 4.0 |
| 66 | Nurses and doctors need to have a good understanding of current evidence-based practice guidelines | 3.7 (1.3) | 3.3, 4.1 | 3.7 (1.2) | 3.1, 4.4 | 3.8 (1.5) | 3.0, 4.6 | 3.5 (1.3) | 2.9, 4.2 |
| 3 | Nurses and doctors need to provide multidisciplinary patient care | 3.6 (1.4) | 3.2, 4.0 | 3.4 (1.5) | 2.5, 4.3 | 3.9 (1.2) | 3.3, 4.5 | 3.4 (1.6) | 2.7, 4.2 |

**Table 2.** *Cont.*

| Number [1] | Statement | All Stakeholders | | Patient | | Nurse | | Doctor | |
|---|---|---|---|---|---|---|---|---|---|
| | | Mean, SD, | 95% CI | Mean, SD, | 95% CI | Mean, SD, | 95% CI | Mean, SD, | 95% CI |
| **Cluster 1, Effective Communication** | | **3.4 (1.3)** | **3.1, 3.8** | **3.5 (1.2)** | **2.8, 4.2** | **3.5 (1.3)** | **2.9, 4.2** | **3.4 (1.2)** | **2.8, 3.9** |
| 7 | Advice from nurses help doctors to plan patient care | 3.6 (1.2) | 3.2, 3.9 | 3.3 (1.4) | 2.5, 4.1 | 3.8 (1.1) | 3.2, 4.4 | 3.5 (1.0) | 3.1, 4.0 |
| 15 | Doctors need to make sure that the instructions they give to nurses is understood | 3.6 (1.1) | 3.4, 3.9 | 3.9 (0.9) | 3.4, 4.4 | 3.4 (1.3) | 2.8, 4.1 | 3.7 (1.1) | 3.2, 4.2 |
| 6 | Nurses and doctors need to trust each other's capabilities | 3.5 (1.2) | 3.1, 3.8 | 3.7 (1.2) | 3.0, 4.5 | 3.4 (1.4) | 2.7, 4.1 | 3.3 (1.1) | 2.8, 3.8 |
| 29 | A structured handover between nurses and doctors is important | 3.4 (1.3) | 3.0, 3.8 | 3.7 (1.0) | 3.1, 4.3 | 3.1 (1.5) | 2.4, 3.9 | 3.4 (1.3) | 2.8, 4.1 |
| 11 | Nurses are a bridge between patient and the doctor | 3.3 (1.5) | 2.9, 3.7 | 3.2 (1.6) | 2.3, 4.1 | 3.9 (1.5) | 3.2, 4.7 | 2.8 (1.3) | 2.2, 3.5 |
| 37 | Nurses and doctors need to make sure that they do not discuss patient care where they can be overheard | 3.2 (1.4) | 2.8, 3.6 | 3.0 (1.4) | 2.1, 3.8 | 3.4 (1.5) | 2.6, 4.2 | 3.0 (1.4) | 2.4, 3.7 |
| 67 | Nurses need prioritize care that impacts patient recovery | 3.2 (1.3) | 2.9, 3.6 | 3.4 (1.4) | 2.5, 4.2 | 3.2 (1.3) | 2.5, 3.8 | 3.2 (1.4) | 2.5, 3.9 |
| 5 | Good nurse–doctor communication reminds clinicians what tasks need to be completed | 3.1 (1.4) | 2.7, 3.5 | 2.6 (1.5) | 1.7, 3.5 | 3.4 (1.1) | 2.8, 3.9 | 3.2 (1.5) | 2.4, 3.9 |
| 25 | Nurses and doctors should discuss care plan before seeing the patient | 3.0 (1.4) | 2.6, 3.4 | 4.0 (0.9) | 3.5, 4.5 | 3.3 (1.3) | 2.6, 4.0 | 1.9 (1.1) | 1.4, 2.5 |
| 45 | Clear allocation of tasks to nurses and doctors | 3.0 (1.1) | 2.7, 3.3 | 3.1 (1.2) | 2.4, 3.8 | 2.7 (1.0) | 2.2, 3.2 | 3.2 (1.2) | 2.6, 3.7 |
| 9 | Clinical problems can only be addressed through positive nurse–doctor communication | 2.9 (1.4) | 2.6, 3.3 | 3.0 (1.4) | 2.2, 3.8 | 3.1 (1.6) | 2.3, 4.0 | 2.8 (1.2) | 2.2, 3.4 |
| 12 | Communication is enhanced if nurses and doctors have consistent shifts (working hours) | 2.2 (1.3) | 1.8, 2.6 | 2.4 (1.5) | 1.6, 3.3 | 1.8 (1.1) | 1.2, 2.4 | 2.4 (1.4) | 1.7, 3.1 |
| **Cluster 2, Trust** | | **3.2 (1.3)** | **2.9, 3.6** | **3.4 (1.2)** | **2.7, 4.1** | **3.3 (1.2)** | **2.7, 4.0** | **3.0 (1.2)** | **2.6, 3.6** |
| 2 | Nurses and doctors need to be good at communicating with family members | 3.9 (1.1) | 3.6, 4.2 | 4.2 (1.0) | 3.6, 4.8 | 4.0 (1.2) | 3.4, 4.6 | 3.5 (1.0) | 3.0, 4.0 |
| 42 | Doctors and nurses need to be honest with patients | 3.9 (1.1) | 3.6, 4.2 | 4.0 (1.3) | 3.3, 4.8 | 3.5 (1.1) | 3.0, 4.1 | 4.0 (1.0) | 3.5, 4.5 |
| 28 | Patients need to fully understand their care and treatment | 3.6 (1.4) | 3.2, 4.0 | 4.4 (0.8) | 3.9, 4.8 | 3.7 (1.5) | 2.9, 4.5 | 2.9 (1.4) | 2.2, 3.6 |

**Table 2.** *Cont.*

| Number [1] | Statement | All Stakeholders | | Patient | | Nurse | | Doctor | |
|---|---|---|---|---|---|---|---|---|---|
| | | Mean, SD, | 95% CI | Mean, SD, | 95% CI | Mean, SD, | 95% CI | Mean, SD, | 95% CI |
| **Cluster 2, Trust** | | **3.2 (1.3)** | **2.9, 3.6** | **3.4 (1.2)** | **2.7, 4.1** | **3.3 (1.2)** | **2.7, 4.0** | **3.0 (1.2)** | **2.6, 3.6** |
| 52 | Good interdisciplinary communication will ensure that discharge plans are meaningful | 3.6 (1.2) | 3.3, 3.9 | 2.9 (1.3) | 2.2, 3.6 | 3.9 (1.1) | 3.3, 4.5 | 3.7 (0.9) | 3.3, 4.1 |
| 40 | Doctors and nurses need to use language that can be understood by the patient | 3.5 (1.4) | 3.1, 3.9 | 4.1 (1.2) | 3.5, 4.8 | 3.7 (1.4) | 2.9, 4.4 | 3.0 (1.5) | 2.3, 3.7 |
| 53 | Good communication between doctors and nurses can comfort patients | 3.3 (1.5) | 2.9, 3.7 | 3.8 (1.4) | 3.0, 4.6 | 3.4 (1.5) | 2.6, 4.1 | 2.9 (1.4) | 2.2, 3.6 |
| 19 | Direct (face-to-face) communication reduces delays in patient care | 3.2 (1.4) | 2.8, 3.6 | 3.2 (1.4) | 2.4, 4.0 | 3.0 (1.2) | 2.4, 3.7 | 3.4 (1.7) | 2.6, 4.2 |
| 1 | Good communication will improve people's faith in medicine | 3.0 (1.3) | 2.7, 3.4 | 3.1 (1.2) | 2.4, 3.8 | 2.8 (1.3) | 2.1, 3.5 | 3.1 (1.3) | 2.5, 3.7 |
| 69 | Patients tend to share more information with nurses than doctors | 2.6 (1.4) | 2.2, 3.0 | 2.2 (1.3) | 1.5, 2.9 | 3.3 (1.6) | 2.5, 4.1 | 2.2 (1.2) | 1.6, 2.8 |
| 38 | Patients can influence communication between nurses and doctors | 2.0 (0.9) | 1.8, 2.3 | 2.1 (1.2) | 1.5, 2.8 | 2.0 (0.8) | 1.6, 2.4 | 2.0 (0.9) | 1.6, 2.5 |
| **Cluster 3, Patient safety** | | **3.1 (1.3)** | **2.8, 3.5** | **3.1 (1.3)** | **2.3, 3.9** | **3.1 (1.3)** | **2.4, 3.9** | **3.1 (1.2)** | **2.5, 3.7** |
| 49 | When vital information is not communicated, it can lead to an increased risk of mortality | 4.2 (1.1) | 3.9, 4.5 | 4.0 (1.1) | 3.4, 4.7 | 3.9 (1.3) | 3.2, 4.5 | 4.6 (0.8) | 4.3, 5.0 |
| 14 | Important information about patient care gets lost if communication is poor | 3.7 (1.2) | 3.3, 4.0 | 3.2 (1.5) | 2.4, 4.1 | 3.7 (1.2) | 3.1, 4.4 | 3.9 (1.0) | 3.5, 4.4 |
| 44 | Poor communication can lead to worse health care outcomes in the longer term | 3.7 (1.2) | 3.4, 4.1 | 3.2 (1.3) | 2.4, 4.0 | 3.6 (1.1) | 3.0, 4.2 | 4.2 (1.1) | 3.6, 4.7 |
| 43 | Bad communication between nurses and doctors may be traumatic for the patient | 3.4 (1.3) | 3.1, 3.8 | 3.5 (1.3) | 2.8, 4.3 | 3.5 (1.3) | 2.9, 4.2 | 3.3 (1.3) | 2.6, 3.9 |
| 48 | Patients can get wrong treatment | 3.2 (1.5) | 2.8, 3.7 | 3.5 (1.5) | 2.6, 4.4 | 3.2 (1.6) | 2.4, 4.0 | 3.1 (1.6) | 2.4, 3.9 |
| 41 | Poor communication may prolong a patient's period of hospitalization | 3.2 (1.4) | 2.8, 3.5 | 3.3 (1.3) | 2.5, 4.0 | 3.2 (1.4) | 2.5, 4.0 | 3.0 (1.4) | 2.3, 3.7 |
| 63 | Delayed communication can lead to frustration | 3.1 (1.3) | 2.8, 3.5 | 3.4 (1.4) | 2.6, 4.2 | 3.2 (1.5) | 2.4, 4.0 | 2.9 (1.2) | 2.3, 3.5 |

**Table 2.** *Cont.*

| Number [1] | Statement | All Stakeholders | | Patient | | Nurse | | Doctor | |
|---|---|---|---|---|---|---|---|---|---|
| | | Mean, SD, | 95% CI | Mean, SD, | 95% CI | Mean, SD, | 95% CI | Mean, SD, | 95% CI |
| **Cluster 3, Patient safety** | | **3.1 (1.3)** | **2.8, 3.5** | **3.1 (1.3)** | **2.3, 3.9** | **3.1 (1.3)** | **2.4, 3.9** | **3.1 (1.2)** | **2.5, 3.7** |
| 36 | Poor communication may mean that patients are sent to an inappropriate clinical setting | 3.1 (1.2) | 2.8, 3.4 | 3.0 (1.1) | 2.4, 3.7 | 3.0 (1.4) | 2.3, 3.8 | 3.2 (1.2) | 2.6, 3.7 |
| 47 | Poor communication may increase the chances of a patient needed to be readmitted | 3.0 (1.4) | 2.7, 3.4 | 3.1 (1.5) | 2.3, 4.0 | 2.9 (1.3) | 2.2, 3.6 | 3.2 (1.4) | 2.5, 3.8 |
| 39 | Poor communication may mean that patients are not clear about the self-care behaviours they need to change | 2.9 (1.4) | 2.5, 3.2 | 3.2 (1.1) | 2.6, 3.9 | 3.0 (1.6) | 2.2, 3.8 | 2.5 (1.3) | 1.9, 3.1 |
| 51 | Poor communication may mean that patients do not get the required interdepartmental consultation on time | 2.9 (1.3) | 2.5, 3.3 | 2.6 (1.4) | 1.8, 3.4 | 3.4 (1.2) | 2.8, 4.0 | 2.7 (1.4) | 2.1, 3.4 |
| 54 | Dissatisfied patients will disengage with healthcare services | 2.8 (1.5) | 2.4, 3.2 | 2.8 (1.6) | 1.9, 3.8 | 2.9 (1.5) | 2.1, 3.7 | 2.7 (1.4) | 2.1, 3.4 |
| 22 | The severity of a patient's condition can impact communication | 2.7 (1.2) | 2.3, 3.0 | 2.3 (1.1) | 1.7, 2.9 | 2.6 (1.4) | 1.9, 3.3 | 3.0 (1.2) | 2.5, 3.6 |
| 50 | Patients can be discharged before they are ready | 2.5 (1.3) | 2.1, 2.9 | 2.5 (1.3) | 1.8, 3.3 | 2.5 (1.4) | 1.8, 3.2 | 2.5 (1.3) | 1.9, 3.1 |
| 58 | Patients are more likely to complain if they witness poor communication between nurses and doctors | 2.5 (1.2) | 2.2, 2.8 | 2.8 (1.1) | 2.2, 3.5 | 2.6 (1.4) | 1.9, 3.4 | 2.2 (1.1) | 1.6, 2.7 |
| **Cluster 4, Impediments to patient care** | | **2.9 (1.2)** | **2.6, 3.2** | **2.9 (1.2)** | **2.2, 3.7** | **2.9 (1.2)** | **2.2, 3.5** | **2.9 (1.1)** | **2.3, 3.5** |
| 60 | Unprofessional conduct (e.g., shouting) between nurses and doctors needs to be reported | 3.8 (1.2) | 3.4, 4.1 | 3.7 (1.4) | 2.8, 4.5 | 3.8 (1.3) | 3.1, 4.5 | 3.8 (1.0) | 3.3, 4.3 |
| 35 | Workplace bullying impacts communication | 3.7 (1.4) | 3.3, 4.1 | 3.6 (1.4) | 2.7, 4.4 | 3.9 (1.4) | 3.2, 4.6 | 3.7 (1.3) | 3.0, 4.3 |
| 57 | Conflict can negatively affect the clinician's wellbeing | 3.3 (1.2) | 3.0, 3.6 | 3.7 (1.0) | 3.2, 4.3 | 2.7 (1.1) | 2.2, 3.3 | 3.4 (1.2) | 2.9, 4.0 |
| 59 | Having English as a second language may impact nurse–doctor communication | 2.8 (1.4) | 2.4, 3.2 | 2.7 (1.6) | 1.8, 3.6 | 3.2 (1.3) | 2.5, 3.9 | 2.5 (1.3) | 1.9, 3.1 |
| 33 | Clinicians with a heavy caseload can be less effective at communicating | 2.8 (1.3) | 2.5, 3.2 | 2.7 (1.2) | 2.0, 3.5 | 2.5 (1.3) | 1.8, 3.2 | 3.2 (1.3) | 2.6, 3.8 |
| 30 | Personal issues (e.g., family stress) can impact communication | 2.7 (1.2) | 2.4, 3.1 | 3.3 (1.3) | 2.5, 4.0 | 2.5 (1.2) | 1.9, 3.1 | 2.5 (0.9) | 2.1, 3.0 |

**Table 2.** *Cont.*

| Number [1] | Statement | All Stakeholders | | Patient | | Nurse | | Doctor | |
|---|---|---|---|---|---|---|---|---|---|
| | | Mean, SD, | 95% CI | Mean, SD, | 95% CI | Mean, SD, | 95% CI | Mean, SD, | 95% CI |
| **Cluster 4, Impediments to patient care** | | **2.9 (1.2)** | **2.6, 3.2** | **2.9 (1.2)** | **2.2, 3.7** | **2.9 (1.2)** | **2.2, 3.5** | **2.9 (1.1)** | **2.3, 3.5** |
| 56 | Poor communication between nurses and doctors may lead to people taking time off work | 2.6 (1.3) | 2.3, 3.0 | 2.8 (1.3) | 2.1, 3.6 | 2.7 (1.4) | 2.0, 3.5 | 2.3 (1.2) | 1.8, 2.9 |
| 62 | Critical comments negatively impact the quality of communication | 2.6 (1.3) | 2.2, 2.9 | 2.7 (1.2) | 2.0, 3.5 | 2.5 (1.5) | 1.8, 3.3 | 2.4 (1.3) | 1.8, 3.1 |
| 34 | Personal Protective Equipment (PPE) is a barrier to effective communication | 1.9 (1.1) | 1.6, 2.2 | 1.2 (0.6) | 0.9, 1.5 | 2.1 (0.9) | 1.7, 2.6 | 2.2 (1.2) | 1.6, 2.8 |
| **Cluster 5, Interpersonal skills** | | **2.7 (1.2)** | **2.3, 3.0** | **2.7 (1.3)** | **1.9, 3.5** | **2.7 (1.2)** | **2.1, 3.3** | **2.6 (1.2)** | **2.0, 3.1** |
| 65 | Effective communication is a skill that needs to be taught when nurses and doctors are in training | 3.9 (1.2) | 3.5, 4.2 | 4.0 (1.0) | 3.4, 4.6 | 3.7 (1.3) | 3.0, 4.3 | 4.0 (1.4) | 3.3, 4.7 |
| 27 | Clinicians need to be approachable | 3.8 (1.2) | 3.4, 4.1 | 3.7 (1.0) | 3.2, 4.3 | 3.5 (1.5) | 2.8, 4.3 | 3.9 (1.1) | 3.4, 4.5 |
| 26 | The quality of communication between nurses and doctors can influence the ward atmosphere | 3.4 (1.3) | 3.0, 3.8 | 3.4 (1.3) | 2.6, 4.1 | 3.3 (1.4) | 2.6, 4.0 | 3.5 (1.3) | 2.9, 4.1 |
| 20 | Orientation of new staff improves effective nurse–doctor communication | 3.0 (1.3) | 2.7, 3.4 | 2.9 (1.5) | 2.0, 3.8 | 3.2 (1.4) | 2.5, 4.0 | 3.0 (1.2) | 2.4, 3.6 |
| 32 | The volume of information shared between nurses and doctors can impact understanding | 3.0 (1.2) | 2.6, 3.3 | 3.4 (1.0) | 2.8, 3.9 | 2.9 (1.2) | 2.3, 3.5 | 2.7 (1.2) | 2.1, 3.3 |
| 64 | Senior clinicians need to proactively help resolve conflicts between nurses and doctors | 2.8 (1.4) | 2.4, 3.2 | 2.7 (1.5) | 1.8, 3.5 | 3.2 (1.2) | 2.6, 3.8 | 2.6 (1.4) | 1.9, 3.3 |
| 24 | Finding time for informal discussions about how to improve patient care is important | 2.7 (1.4) | 2.3, 3.1 | 2.7 (1.5) | 1.8, 3.5 | 3.4 (1.4) | 2.6, 4.1 | 2.0 (1.2) | 1.5, 2.6 |
| 55 | Technology can be used to improve communication between nurses and doctors | 2.7 (1.4) | 2.3, 3.1 | 2.8 (1.4) | 2.0, 3.6 | 2.6 (1.6) | 1.8, 3.4 | 2.7 (1.2) | 2.1, 3.3 |
| 18 | Using clinicians' name in discussions improves communication | 2.5 (1.4) | 2.1, 2.9 | 2.4 (1.6) | 1.4, 3.3 | 2.9 (1.3) | 2.2, 3.6 | 2.3 (1.2) | 1.7, 2.9 |
| 21 | Communication is improved if nurses and doctors spend time getting to know each other | 2.3 (1.2) | 2.0, 2.7 | 2.5 (1.2) | 1.8, 3.2 | 2.4 (1.3) | 1.8, 3.1 | 2.0 (1.2) | 1.5, 2.6 |

**Table 2.** *Cont.*

| Number [1] | Statement | All Stakeholders | | Patient | | Nurse | | Doctor | |
|---|---|---|---|---|---|---|---|---|---|
| | | Mean, SD, | 95% CI | Mean, SD, | 95% CI | Mean, SD, | 95% CI | Mean, SD, | 95% CI |
| **Cluster 5, Interpersonal skills** | | **2.7 (1.2)** | **2.3, 3.0** | **2.7 (1.3)** | **1.9, 3.5** | **2.7 (1.2)** | **2.1, 3.3** | **2.6 (1.2)** | **2.0, 3.1** |
| 10 | Clinicians have a different scope of practice | 2.2 (1.4) | 1.9, 2.6 | 1.9 (1.3) | 1.1, 2.7 | 2.2 (1.4) | 1.5, 2.9 | 2.5 (1.4) | 1.9, 3.2 |
| 46 | Doctors' use of medical jargon impacts understanding by nurses | 2.1 (1.2) | 1.8, 2.5 | 2.4 (1.5) | 1.5, 3.2 | 2.0 (1.1) | 1.4, 2.6 | 2.0 (1.2) | 1.5, 2.6 |
| 68 | Doctors need to lead nurse–doctor communication | 2.0 (1.3) | 1.6, 2.3 | 2.2 (1.6) | 1.3, 3.1 | 1.6 (1.0) | 1.1, 2.1 | 2.2 (1.3) | 1.6, 2.8 |
| 31 | Clinicians with more clinical experience are better at communicating | 1.9 (1.1) | 1.6, 2.2 | 2.1 (1.5) | 1.3, 3.0 | 1.8 (1.0) | 1.3, 2.3 | 1.9 (1.0) | 1.4, 2.4 |
| 16 | Nurses need to lead nurse–doctor communication | 1.7 (1.0) | 1.5, 2.0 | 1.6 (0.8) | 1.1, 2.1 | 2.0 (0.9) | 1.5, 2.4 | 1.6 (1.1) | 1.1, 2.1 |

[1] Number indicates the sequence (generated using the random number generation function of Microsoft Excel) that was entered in the Ariadne software.

### 3.3. Phase 4, Representation of the Statements

Figure 2 is the point map produced from the clustering data, locating each statement (as a dot) in a two-dimensional space. Seventeen candidate concept maps were subsequently generated by Ariadne software and can be accessed as Supplementary Document S6.

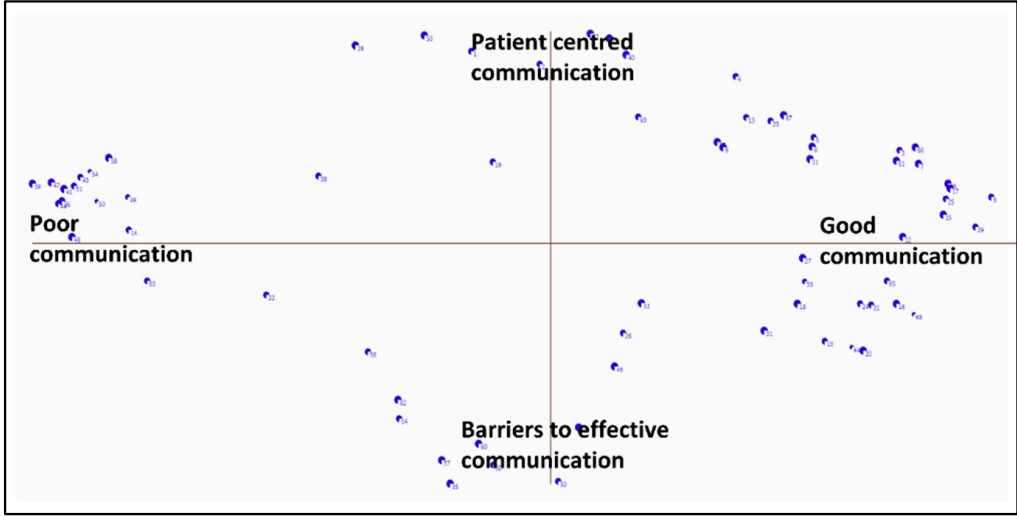

**Figure 2.** Point map showing the location of individual statements against the *x* and *y* axes.

### 3.4. Phase 5, Data Interpretation

The research team reviewed each candidate concept map (numbered 2 through 18 in Supplementary Document S6) and selected 6 (5 to 10 cluster solutions) to take forward to the stakeholder interpretation workshop. Concept maps two to four were not considered for further discussion, as many statements were not coherent. We removed concept maps 11 through 18, as multiple clusters comprised a single statement.

Five stakeholders (two patients, two nurses, and one doctor) participated in the data interpretation workshop facilitated by one of the members of the research team (RG). The group reviewed each of the six candidate concept maps, considering the strengths and

limitations of each solution. The stakeholder group considered the five-cluster concept map as their preferred solution that best reflected the data. During the workshop, participants also suggested possible labels for each cluster. However, participants were not able to finalize labels due to time restrictions. The research team made the final decision about the final cluster labels which was a deviation from the planned protocol (Pantha et al., 2021) [21]. Figure 3 is the final concept map.

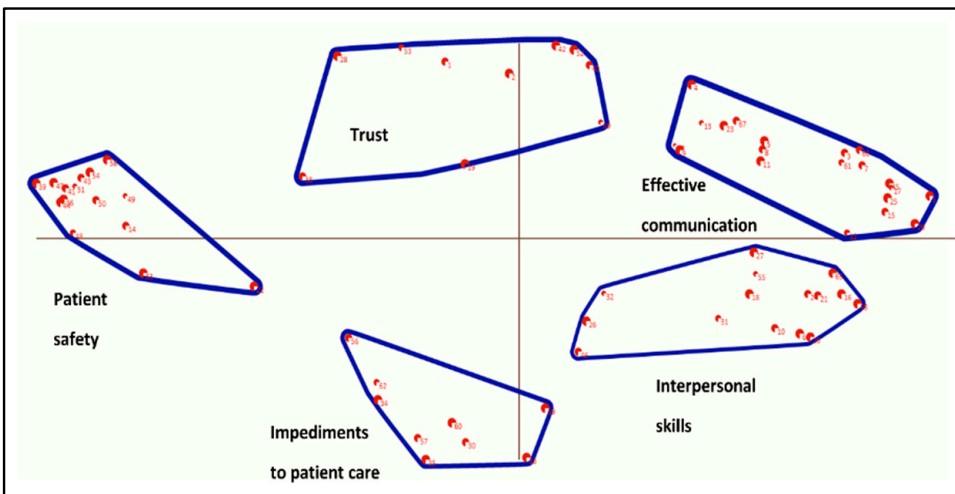

**Figure 3.** Five-cluster concept map generated by the participants.

### 3.4.1. Description of the Axes

Concept maps locate individual statements and clusters on *x*- and *y*-axes (Figure 2). The statements on the *x*-axis represent the quality of communication between nurses and doctors. Statements and clusters located toward the west suggest poor communication while those toward the east suggest good communication. The *y*-axis reflects two concepts labelled "patient-centered communication" located toward the north of the map and "barriers to effective communication" located on the south.

### 3.4.2. Description of the Clusters

Figure 3 shows the final concept map with five clusters that circle the center of the map. The cluster rankings for all participants and by individual stakeholder group are shown in Table 2. The importance rating for all five clusters is toward the middle of the five-point scale. Figure 4 is a pattern match (or ladder) graph showing the rank order for the five clusters. There were no important differences between stakeholder groups as to which clusters were considered more or less important.

### 3.4.3. Cluster 1, Effective Communication

"Effective communication" was the cluster rated as most important and included 20 statements. The cluster captured the importance of good communication between doctors and nurses to ensure high-quality, timely patient care. The cluster was located toward the north and extreme east of the concept map. Example statements include "Precise communication is needed in emergency situations", "A structured handover between nurses and doctors is important", and "Effective nurse-doctor communication ensures timely patient care".

### 3.4.4. Cluster 2, Trust

Containing ten statements, the "Trust" cluster was in the centre and toward the north edge of the concept map. Statements in this cluster include "Doctors and nurses need to be honest with patients", "Patient needs to fully understand their care and treatment", and "Good communication will improve people's faith in medicine".

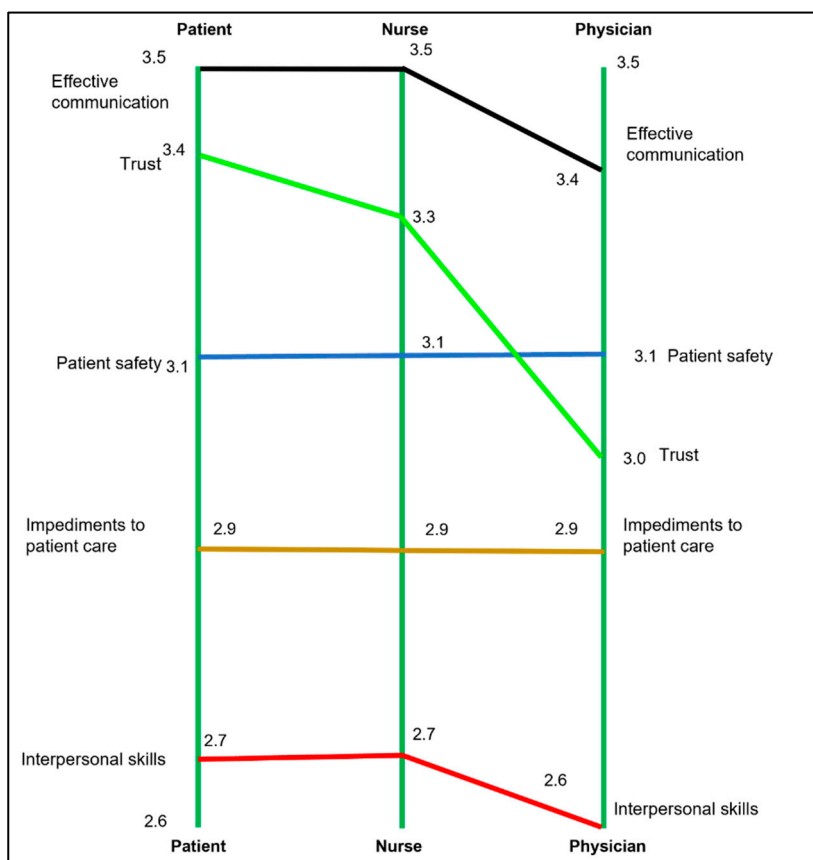

**Figure 4.** Pattern match showing stakeholders' rankings of five clusters.

### 3.4.5. Cluster 3, Patient Safety

The "Patient safety" cluster had 15 statements. This cluster describes how patient safety outcomes can be impacted by poor nurse–doctor communication. Example statements include "Important information about patient care gets lost if communication is poor", "Poor communication can lead to worse health care outcomes in the longer term", and "Dissatisfied patients will disengage with healthcare services". The cluster was located on the extreme west of the concept map across the *x*-axis.

### 3.4.6. Cluster 4, Impediments to Patient Care

There were nine statements in the cluster "Impediments to patient care". The cluster reflects workplace factors that can potentially impact patient care. The cluster was located centrally toward the extreme south of the map. Statements in the cluster include "Workplace bullying impacts communication", "Clinicians with heavy caseload can be less effective at communicating", and "Critical comments negatively impact the quality of communication".

### 3.4.7. Cluster 5, Interpersonal Skills

Occupying the southeast quadrant of the concept map the "Interpersonal skills" cluster includes 15 statements. The cluster encompasses interpersonal strategies that enhance nurse–doctor communication and include "Clinicians with more clinical experience are better at communicating", "Finding time for informal discussions about patient care is important", and "Using clinicians name in discussion improves communication".

## 4. Discussion

This study used concept mapping to examine stakeholder perspectives on how nurse–doctor communication impacts patient care. Stakeholder groups identified that patient care was affected by the individual (e.g., clinical experience), interpersonal (e.g., trust), and work-

place aspects (e.g., consistent shifts) of nurse–doctor communication. Broadly, all aspects of nurse–doctor communication were considered of equal importance by study participants.

The concepts identified in our study are consistent with those reported in previous reviews of nurse–doctor communication [27–33]. For example, a review of 38 studies by Bookey-Bassett et al. (2017) identified that effective communication and trust between clinicians facilitated enhanced management of long-term conditions. Similarly, Stutsky and Laschinger (2014), in their review of interprofessional communication, found that trust, patient safety, and interpersonal skills were important in providing high-quality patient care.

Three features of our concept map require consideration. First, clusters are comparatively large but of approximately equal size, underscoring the observation that no one concept is more important than any other in terms of how nurse–doctor communication impacts patient care. The sizes of the clusters could be due to important differences in how participants organized statements into groups during the clustering task. Second, statements within the clusters are not located close to one another, suggesting clusters may lack a clear conceptual focus [24], underscoring the complexities of how patient care is impacted by nurse–doctor communication. Authors of reviews of interprofessional communication have also highlighted the complexity of interdisciplinary working [32–34]. Potentially, our findings may explain why there are several examples in the literature where authors have focused on targeting a single aspect of nurse–doctor communication (e.g., joint clinical rounding, shared care planning) and failed to show improvements in patient outcomes [35,36]. Finally, two of the clusters (effective communication and interpersonal skills) are particularly close to each other, suggesting that they may be part of a single larger concept.

In this study, we included people with recent lived experiences of using inpatient health services. This extends previous research that has exclusively tended to focus on the views of clinicians [2,8,32].

Our study suggests that nurse–doctor communication is multi-dimensional. The five concepts formed a circle around the centre of the map, suggesting that each should be considered equally important to interprofessional communication. We need to acknowledge the complexity of communication when developing strategies to enhance interprofessional working. For example, simulation-based interprofessional trainings to enhance clinical collaboration could incorporate a discussion on the five elements identified in the concept map, such as the impact of communication on patient safety.

## 5. Limitations

There are several important limitations to consider when interpreting the findings of this study. First, participants were recruited through specific social media groups. It is plausible that there are important differences between people that do and do not engage with social media [37], potentially introducing selection bias. Around one in four participants did not complete the prioritizing and clustering tasks. However, our attrition rate was seemingly typical of concept mapping studies [20], and there did not appear to be a systematic reason for people dropping out of the research. We also note that concept mapping is a group process, and the final concept map is the reflection of the work of all participants rather than individuals per se. Concept maps produced by patients, nurses, and doctors were not analyzed separately. That said, we did not observe important differences between how the three stakeholders ranked individual statements and clusters. Finally, the axis and cluster labels were determined by the research team and not by study participants as is recommended in concept mapping [24]. Whilst stakeholders proposed cluster labels due to time constraints, consensus could not be reached. We acknowledge that the stakeholders could have labelled the clusters differently.

## 6. Conclusions

Patients, nurses, and doctors had broadly consistent views on how nurse–doctor communication may impact patient care. The relatively equal size and orientation of the clusters may suggest that all clusters are broadly of equal importance. There is a need for further research to develop insights into how patient care is influenced by clinical communication.

**Supplementary Materials:** The following supporting information can be downloaded at: https://www.mdpi.com/article/10.3390/nursrep13040133/s1. Supplementary Document S1 social media advert used for the study, Supplementary Document S2, complete list of all statements generated from the study, Supplementary Document S3, list of statements after first round of the reduction, Supplementary Document S4, list of statements removed during the second round of statement reduction, Supplementary Document S5, final list of 69 statements, Supplementary Document S6, 17 candidate concept maps generated using Ariadne software.

**Author Contributions:** All authors were involved in the conception of the study and development of the focus prompt. S.P. conducted brainstorming interviews, extracted statements from the interviews, and undertook the first and second rounds of statement reduction. All authors were involved in the final stages of statement reduction. S.P. facilitated communication with participants for rating and clustering tasks and analyzed Ariadne data to generate candidate concept maps. Maps were reviewed and shortlisted by all authors. R.G. facilitated the stakeholder meeting for data interpretation. S.P. prepared the first draft of the manuscript. R.G. undertook an extensive revision of the first draft of the manuscript. All authors have read and agreed to the published version of the manuscript.

**Funding:** The corresponding author has received a La Trobe University Postgraduate Research Scholarship (LTUPRS) and La Trobe University Full Fee Research Scholarship (LTUFFRS). There was no additional funding for the research. Participants were reimbursed a digital voucher of 25 AUD for each phase of the study. Fund for the reimbursement was supported by the HDR research grants for PhD study.

**Institutional Review Board Statement:** The study was approved by the La Trobe University human ethics committee (HEC–20172).

**Informed Consent Statement:** All participants provided a written informed consent before the brainstorming interview. At each phase of the research, participants were asked if they still wanted to take part in the research.

**Data Availability Statement:** All data related to the research have been provided as supplementary materials (Supplementary Documents S1–S6).

**Public Involvement Statement:** Concept mapping is a participatory research method that involves the public in the work from conception through interpretation and dissemination.

**Guidelines and Standards Statement:** We searched the EQUATOR network website for the availability of a reporting guideline for concept mapping research. There is no reporting guideline available for the reporting of concept mapping research.

**Conflicts of Interest:** Richard Gray is the editor-in-chief of nursing reports.

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
