# Peer review of "Stakeholders’ Perceptions of How Nurse–Doctor Communication Impacts Patient Care: A Concept Mapping Study"

_nursrep, doi:10.3390/nursrep13040133_

Round 1

Reviewer 1 Report

Comments and Suggestions for Authors

Dear Authors,

The manuscript is well-written.  The impact of nurse-doctor communication on patient care is significant, and it affects the care of patients in all aspects.   However, the manner in which the sample was chosen influenced your results.  Doctors, nurses, and patients were not drawn from the same hospitals or clinics.  The ecological fallacy could be a significant bias here.

How can we benefit from your findings in the healthcare system to give improved patient care? Do you have any practical suggestions? 

Author Response

Comment 1

The manuscript is well-written.  The impact of nurse-doctor communication on patient care is significant, and it affects the care of patients in all aspects.

However, the manner in which the sample was chosen influenced your results.  Doctors, nurses, and patients were not drawn from the same hospitals or clinics. The ecological fallacy could be a significant bias here.

Our response:

We do not believe the ecological fallacy applies to our concept mapping research. In concept mapping people with similar characteristics (stakeholders) to take part in developing an understanding of the concept. We are not statistically inferring anything to make a difference based on the population.

We argue that participants from different hospital may add strength to our research as people from different settings were able to participate and share their views. 

Comment 2

How can we benefit from your findings in the healthcare system to give improved patient care?

Our response

Our study adds to the understanding of clinical communication. We have explained in the discussion that our study suggests nurse doctor communication is multidimensional and equally influenced by five concepts identified in the sutdy. We have reflected this in our manuscript as:

Our study suggests that nurse-doctor communication is multi-dimensional. The five concepts formed a circle around the centre of the map, suggesting that each should be considered equally important to interprofessional communication.  (lines 372-375)

Comment 3

Do you have any practical suggestions? 

Our response:

We have added a section in the discussion to address the query.

We need to acknowledge the complexity of communication when developing strategies to enhance interprofessional working. For example, simulation-based interprofessional trainings to enhance clinical collaboration could incorporate a discussion on the five elements identified in the concept map, such as, the impact of communication on patient safety. (lines 376-380)

Reviewer 2 Report

Comments and Suggestions for Authors

The authors have identified an important aspect of healthcare delivery, that is nurse-doctor communication. They are to be commended in further identifying that nurse-doctor communication in conversations with healthcare professionals is a prerequisite for patient care. The manuscript contains minor issues which need considering. I would like to add the following remarks.

1. Although nurse-doctor communication is appealing as a concept and should translate to benefit, in practice there is little real evidence for this, and the research needed to demonstrate the association is difficult.

2. In most parts of the world nurse-doctor communication is very patchily evident, and indeed even within hospitals and departments peoples' experience of this can be very variable; this is, after all, a manifestation of leadership styles. In many parts of the world, healthcare is still extremely hierarchical and cultural norms.

Author Response

Thank you for the feedback. Below is the point by point response to the feedback and comments from the reviewer.

The authors have identified an important aspect of healthcare delivery, that is nurse-doctor communication. They are to be commended in further identifying that nurse-doctor communication in conversations with healthcare professionals is a prerequisite for patient care. The manuscript contains minor issues which need considering. I would like to add the following remarks.

Remark 1

Although nurse-doctor communication is appealing as a concept and should translate to benefit, in practice there is little real evidence for this, and the research needed to demonstrate the association is difficult.

Our response

We acknowledge the reviewers’ remarks that there is little real evidence to demonstrate how clinical communication impacts patient care. As communication is influenced by various factors, further research is needed to demonstrate the association.

Remark 2

In most parts of the world nurse-doctor communication is very patchily evident, and indeed even within hospitals and departments peoples' experience of this can be very variable; this is, after all, a manifestation of leadership styles. In many parts of the world, healthcare is still extremely hierarchical and cultural norms.

Our response
We acknowledge to the influence of various personal and organizational factors on clinical communication.